# Assessment of Differences in the Dimensions of Mandible Condyle Models in Fan- versus Cone-Beam Computer Tomography Acquisition

**DOI:** 10.3390/ma14061388

**Published:** 2021-03-12

**Authors:** Bartosz Bielecki-Kowalski, Marcin Kozakiewicz

**Affiliations:** Department of Maxillofacial Surgery, Medical University of Lodz, 113 Żeromskiego str, 90-549 Lodz, Poland; marcin.kozakiewicz@umed.lodz.pl

**Keywords:** mandible condyle, anatomy, fan-beam computed tomography, cone-beam computed tomography, radiological modeling, CAD/CAM, segmentation

## Abstract

Modern treatment in the field of head and neck surgery aims for the least invasive therapy and places great emphasis on restorative treatment, especially in the case of injury and deformation corrective surgery. More and more often, surgeons use CAD/CAM (Computer-Aided Design and Computer-Aided Manufacturing) tools in their daily practice in the form of models, templates, and computer simulations of planning. These tools are based on DICOM (Digital Imaging and Communications in Medicine) files derived from computed tomography. They can be obtained from both fan-beam (FBCT) and cone-beam tomography (CBCT) acquisitions, which are subsequently segmented in order to transform them into a 1-bit 3D model, which is the basis for further CAD processes. Aim: Evaluation of differences in the dimensions of mandible condyle models in fan- versus cone-beam computer tomography for surgical treatment purposes. Methods: 499 healthy condyles were examined in CT-based 3D models of Caucasians aged 8–88 years old. Datasets were obtained from 66 CBCT and 184 FBCT axial image series (in each case, imaging both mandible condyles resulted in the acquisition of 132 condyles from CBCT and 368 condyles from FBCT) and were transformed into three-dimensional models by digital segmentation. Eleven different measurements were performed to obtain information whether there were any differences between FBCT and CBCT models of the same anatomical region. Results: 7 of 11 dimensions were significantly higher in FBCT versus lower in CBCT (*p* < 0.05).

## 1. Introduction

Currently, computer tomography (CT) scanning is widely described as the golden standard of imaging techniques of the head and neck [1]. The accurate, constantly improved resolution of CT allows surgical procedures to be planned more accurately. Modern surgical treatment of head and neck diseases aims at performing the least invasive procedures and places great emphasis on reconstructive treatment, especially in traumatology and reconstructive surgery [2]. More and more often, surgeons use CAD/CAM tools in the form of models, templates, and computer simulations before performing procedures [3,4,5]. Medical files obtained during computed tomography (DICOM) are the basis on which these tools are created [6].

The last decade has seen widespread use of a new technique in the acquisition of tomographic images: cone-beam computed tomography. It is used in traumatology [7], orthognathic surgery [8], temporomandibular joint disorders [9], infections treatment [10], and tooth and cyst removal [11]. Radiological modality is an intriguing tool for treatment planning in implantologic treatment [12], prosthodontics [13], and orthodontic treatment [14], but it is not without its disadvantages [15].

DICOM files can be obtained both from fan-beam and cone-beam tomography studies. They subsequently undergo a segmentation process which transforms them into a 1-bit 3D model, which makes up the foundation for further CAD processes. Both of these types of tomography are an excellent tool for scientists conducting anthropometric research [16].

The aim of the study was to compare anatomical measurements of the mandibular condylar region obtained from the fan-beam tomography and cone-beam tomography in the context of their suitability for planning further surgical treatment.

## 2. Materials and Methods

Approvals from the bioethics committee at the Medical University of Lodz (numbers: RNN/125/15/KE and RNN/738/12/KB) were obtained for the study. The DICOM files were found in the databases of medical institutions. The authors had no direct contact with humans. The study dealt exclusively with the transformation of computer files.

Four hundred ninety-nine mandibular condyles were examined in CT-based 3D models of Caucasians aged 8–88. Datasets were obtained from CBCT using a Carestream CS 9300 3D scanner (Carestream Dental LLC, Atlanta, GA, USA) and a 320-MDCT (Multidetector Computed Tomography) volumetric scanner (Aquilion ONE, Toshiba, Otawara, Japan). Sixty-six CBCT and 184 MDCT images (imaging both mandible condyles resulted in acquisition of 132 condyles from CBCT and 368 condyles from FBCT) were acquired after a procedure of anonymization [17] from the Maxillofacial Surgery Clinic Database. The FoV (Field of View) of CBCT images was 17 × 13.5 cm. Images of patients suffering from degenerative lesions in the temporomandibular joint (TMJ) region (for example, ankylosis) and modeling changes in the mandibula (tumor growth, dysplasia) were excluded. The following were also excluded: post-traumatic tomography images in the region of the mandible, after open rigid internal fixation (ORIF) and after resection of mandible patients, low quality tomographic images, and numerous artifacts. DICOM axial image series were transformed into three-dimensional models of bone-use segmentation. Bone segmentation was performed using global thresholding defined for the CBCT and the FBCT by individual histogram analysis according to Baillard and Barillot’s protocol [18]. Subsequently obtained models were subjected to measurements. Segmentation, model preparation, and measurements were performed in Mimics 17.0 software (Materialise, Leuven, Belgium). The mandibular bone was semiautomatically delineated by using a global threshold algorithm. The computer-suggested bone threshold values were visually confirmed in order to allow for the best segmentation overlap with the original image of the condyles. For all models, the posterior ramus line (base line) was determined based on the algorithm described by Neff [19] (Table 1; Figure 1).

Statistical analysis was performed in Statgraphics Centurion Version 18.1.12 (StarPoint Technologies. INC., Falls Church, VA, USA). The relation of categorical data was tested by the χ^2^ independence test and quantitative data was analyzed by ANOVA as a detected normal distribution with stable variance or by the Kruskal–Wallis test. The significance level was established as *p* < 0.05.

## 3. Results

A total of 499 models of the condylar process were obtained. Three hundred sixty-seven models from the fan-beam tomography and 132 models from the cone-beam tomography were created. Seventy-two women and 60 men were examined with the cone-beam tomography, and 110 women and 257 men were examined with the fan-beam tomography. The χ^2^ test of independence showed that more men than women were tested statistically (*p* < 0.05). The patients tested with CBCT were of the same age as the patients tested with fan-beam tomography (*p* > 0.05). The median age of the patients examined with CBCT was 40 ± 14.5 and patients examined with fan-beam tomography was 41 ± 18.9 (Figure 2).

Among the measurements analyzed (Table 2) with the Kruskal–Wallis test, the distance_sigmoidnotch-necktop measurement did not show statistically significant differences between the models generated on the basis of cone-beam tomography and fan-beam tomography. Statistically significant differences between the two types of tomography were obtained for the length_neck_top, width_neck_basal, and thickness_sigmoid_notch measurements (*p* < 0.05).

For the measurements analyzed with the ANOVA test, no statistically significant differences were found depending on the type of tomography for the measurements: height_neck, height_neck_new_classification, and the condyle angulation. However, statistically significant differences were found for the measurements of the Ramus height, width_head, length_neck_middle, and length_neck_basal (*p* < 0.05) (Figure 3).

## 4. Discussion

### 4.1. Differences in CBCT and FBCT Segmentation

Segmentation is the process of converting a multi-bit computed tomography image into a single-bit 3D model. Its accuracy depends directly on the resolution of the tomography [20,21]. The noticeably greater contrast between soft tissue and bone makes FBCT easier to segment than CBCT [22]. This conclusion is consistent with the results obtained in our research. Differences in CBCT segmentation also result from a variety of settings, particularly the voxel resolution [20,23]. There are still only a few studies comparing the accuracy of CBCT and FBCT segmentation performed in vivo [24]. Tests performed on dry skulls or cadavers do not fully reflect the problem of segmentation of CT examinations obtained from living humans [22]. 

Images obtained from FBCT segmentation are characterized by a more accurate representation of the compact bone structure; however, this examination is more susceptible to the appearance of artifacts, e.g., resulting from the presence of metal prosthetics in the oral cavity [24,25]. This can make it difficult to obtain accurate measurements in the dental area.

Vandenberghe et al. compared models of toothless alveolar processes of the maxilla and the mandible. They found similar accuracy for the models obtained from the segmentation of CBCT and FBCT. However, they drew attention to the differences between the models obtained from the CBCT studies. The differences were revealed especially when using different exposition times and voxel resolutions [26]. Hassan et al. noted that the quality of the models created as a result of CBCT segmentation was significantly influenced by the field of view (FoV) applied [22]. A smaller imaging field is characterized by greater accuracy of the obtained images. In maxillofacial traumatology, it is necessary to use the highest possible FoV to visualize the entire facial skeleton, which may translate the results into a decreased accuracy for CBCT-based diagnoses. Our CBCT FoVs were relatively large (17 × 13.5 cm), which may be one of the probable reasons for the differences in the measurements obtained.

The studies conducted by Wang et al. [27] allowed for the obtainment of a higher accuracy of mapping anatomical structures because of the adoption of the random forests method [28]. They obtained a CBCT and FBCT image segmentation protocol. However, this study had some limitations, which include: a small group size and a susceptibility to the presence of metal artifacts.

Kainmueller et al. studied the accuracy of segmentation of the inferior alveolar canal on the basis of 105 CBCTs. They noticed that the largest errors in segmentation arose in the area of mental protuberance and the condyles [29]. Gollmer et al. arrived at a similar conclusion in their work while examining 30 models of the lower jaw based on CBCT segmentation [30]. The proposed explanation is related with the fringe of the field of view, which is associated with greater susceptibility to interference. The results obtained by the authors are consistent with the results obtained in our work.

### 4.2. Anatomical Measurements in CBCT- and FBCT-Based Models

Some authors agree that CBCT underestimates the measurement results [1,31], as was noticed in this study. Lascala et al., comparing the linear measurements obtained in CBCT-based models with the actual measurements of dry skulls, concluded that CBCT images underestimate the real distances [32], which is consistent with the results of our research in the regions of the condyle neck base, the middle neck, the top neck, the sigmoid notch, and the mandible head. The differences are especially visible in the craniofacial regions where the bone is very thin (e.g., the sigmoid notch area of mandible) [31]. Our study confirms such a relationship. This can be explained by the partial volume effect (PVE) based on the estimation of the gray level of individual voxels, which, in the case of CBCT images, may lead to an incorrect assignment of voxels of the cortical and spongy bone and, in consequence, may change the final image [23,33].

Gomes et al. compared images of the condylar processes and mandibular heads obtained from FBCT and CBCT segmentation in patients qualified for condylar resection and prosthetic joint replacement [34]. The authors obtained similar results comparing 4002 correspondent surface mesh points of models obtained from CBCT and FBCT segmentation. They emphasized that the differences were <1 mm. However, it is noteworthy that the authors did not compare areas of the thin cortex-like sigmoid notch.

### 4.3. Clinical Implications

In the practice of an oral surgeon, the results obtained may affect, for example, the rigidity of the fracture fixation performed. Following the image of the cone-beam tomography in the area of the thin bone, the surgeon may decide to use shorter screws for fracture osteosynthesis to avoid through and through osteosynthesis. The use of a shorter screw adversely affects the pull-out force of the miniplate [35,36] and the screws [34,37,38].

## 5. Conclusions

The indisputable advantage of cone-beam computer tomography is the lower dose of radiation taken by patients during the examination. CBCT parameters such as FoV, voxel width, and exposure time should be taken into account and special care should be taken when using it as a basis for treatment planning in the region of thin bone and in cases where accuracy less than 1 mm is indicated. This is because of the undersizing of anatomical elements in comparison with fan-beam computer tomography.

## Figures and Tables

**Figure 1 materials-14-01388-f001:**
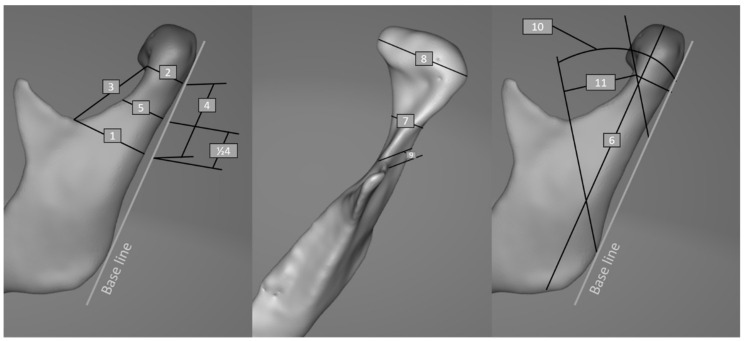
Anatomical measurements: (1) length_neck_basal, (2) length_neck_top, (3) distance_sigmoidnotch-neck top, (4) height_neck, (5) length_neck_middle, (6) the Ramus height, (7) width_neck_basal, (8) width_head, (9) thickness_sigmoid_notch, (10) angle_posteriorline-notchpoint, (11) high_neck_new_classification.

**Figure 2 materials-14-01388-f002:**
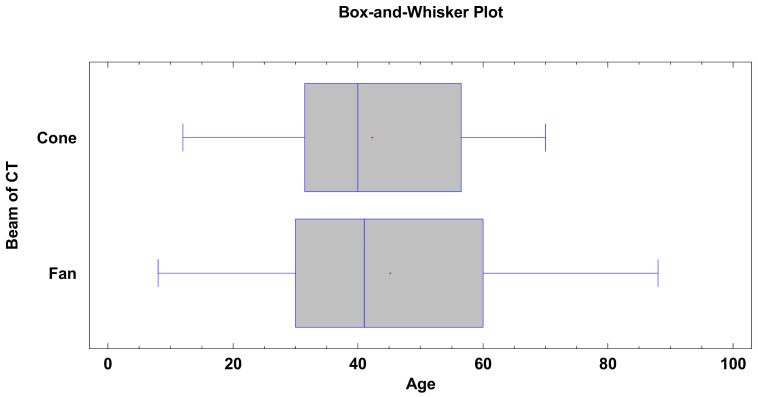
Results showing statistically insignificant (*p* > 0.05) differences between the age of patients and the type of computer tomography (CT) subjected to segmentation.

**Figure 3 materials-14-01388-f003:**
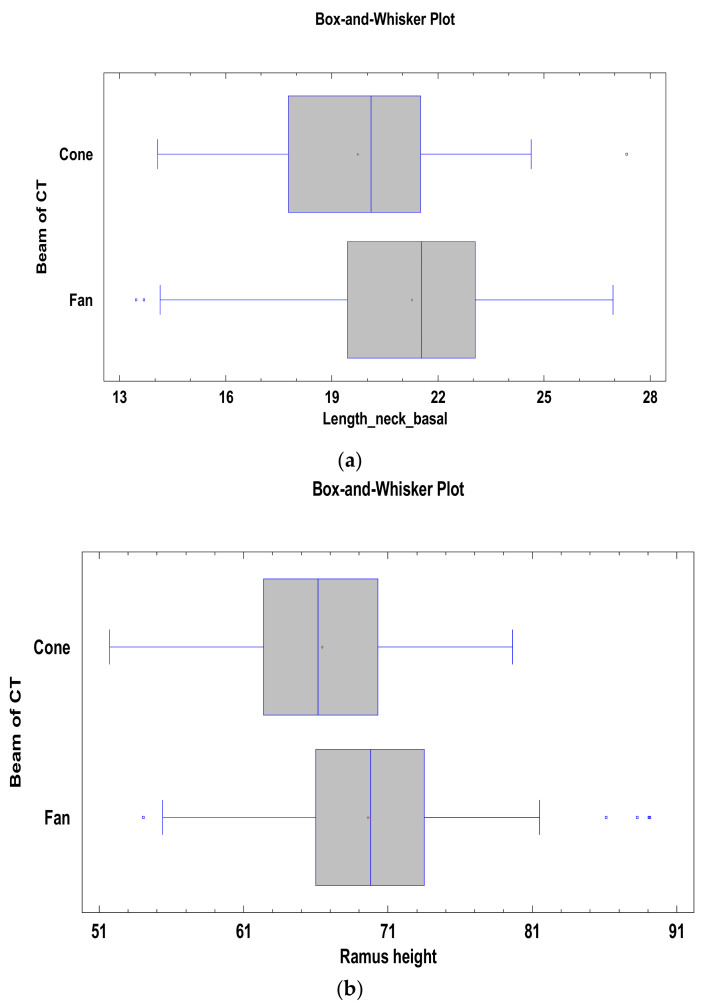
Results showing statistically significant (*p* < 0.05) differences between measurements: Length_neck_basal (**a**), Ramus height (**b**), Length_neck_middle (**c**), Thickness_sigmoid_notch (**d**), Width_head (**e**), Length_neck_top (**f**), Width_neck_basal (**g**) depending on the CT type subjected to segmentation.

**Table 1 materials-14-01388-t001:** Methods for application of anatomical measurements.

Measurement	Starting Point of Measurement	End Point of Measurement	Comment
Length_neck_basal	Lowest point in semilunar notch	Most backward point at point of semilunar notch	Measured perpendicularly to the baseline
Length_neck_top	Most forward point at the level of the condylar head reference line described by Neff [19]	Most backward point at the level of the condylar head reference line described by Neff [19]	Measured perpendicularly to the baseline
Distance_sigmoidnotch-necktop	Most forward point of length_neck_basal	Most forward point of length_neck_top	
Height_neck	Length_neck_basal	Length_neck_top	Measured parallel to the baseline
Length_neck_middle	Most forward point at the level of ½ of height_neck	Most backward point at the level of ½ of height_neck	Measured perpendicularly to the baseline
Ramus height	Lowest point of the Ramus height	The highest point of the Ramus height	Measured parallelly to the baseline
Width_neck_basal	Most mesial point at level of length_neck_basal	Most distal point at level of length_neck_basal	Measured in frontal projection perpendicularly to the baseline
Width_head	Variable	Variable	The widest measurement of the head of the mandible measured in the frontal projection perpendicularly to the baseline
Thickness_sigmoid_notch	Variable	Variable	Width measured 1 mm below the sigmoid notch measured in the frontal projection perpendicularly to the baseline
Angle_posteriorline-notchpoint			Angle between the lowest point in the semilunar notch, the lowest point on the baseline line, and the baseline line itself
Height_neck_new_classification	Medial arm of angle_posteriorline-notchpoint	Medial arm of the angle carried out by the most forward points of length_neck_top	Corresponding angles

**Table 2 materials-14-01388-t002:** Obtained measurement results of the condylar process of the mandible in the two radiological imaging techniques.

Measurement Names	FBCT Mean ± SD	CBCT Mean ± SD	Statistical Significance
Length_neck_basal	21.27 ± 2.57	19.74 ± 2.45	*p* < 0.05
Length_neck_top	12.07 ± 1.79	11.21 ± 1.47	*p* < 0.05
Distance_sigmoidnotch-necktop	14.91 ± 3.16	14.3 ± 2.76	n.s.
Height_neck	10.26 ± 2.8	10.19 ± 2.59	n.s.
Length_neck_middle	13.18 ± 1.89	12.33 ± 1.66	*p* < 0.05
Ramus height	69.61 ± 5.93	66.44 ± 5.46	*p* < 0.05
Width_neck_basal	10.29 ± 3.13	9.22 ± 2.46	*p* < 0.05
Width_head	20.74 ± 2.44	19.64 ± 2.45	*p* < 0.05
Thickness_sigmoid_notch	2.15 ± 0.81	1.71 ± 0.53	*p* < 0.05
Angle_posteriorline-notchpoint	37.95 ± 18.77	34.54 ± 18.64	n.s.
Height_neck_new_classification	14.56 ± 2.88	13.94 ± 2.79	n.s.

FBCT—fan-beam computer tomography; CBCT—cone-beam computer tomography; SD—standard deviation.

## Data Availability

Data sharing is not applicable to this article.

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
