# Peer review of "Assessment of Differences in the Dimensions of Mandible Condyle Models in Fan- versus Cone-Beam Computer Tomography Acquisition"

_materials, 2021, doi:10.3390/ma14061388_

Round 1

Reviewer 1 Report

  1. There are some inadvertences in the text:
  • In the Abstract it is mentioned that of 499 healthy condyles examined, 66 were CBCT and 184 were FBCT axial image series.
  • In the Materials and Methods section the authors mention 66 CBCT and 184 MDCT images
  • In the Results section, the authors state a number of 499 models of the condylar process that were obtained, of which 367 models from fan beam tomography (FBCT) and 132 models from cone beam tomography (CBCT)

*Please consider a more organized view on the number and types of images processed

  1. A table should be considered for the legend of Figure 1 explained on rows 80-98.
  2. Numbers 10, 11 represented on the figure don’t have a description
  3. Reference 38 noted on row 195 does not appear in the references section.
  4. The conclusion section is formulated in a vague manner; conclusions should reinforce the central idea of the paper.

Author Response

  1. There are some inadvertences in the text:
  • In the Abstract it is mentioned that of 499 healthy condyles examined, 66 were CBCT and 184 were FBCT axial image series.
  • The number is due to the fact that each CT scan provided 2 mandibular condylar processes (right and left). It has been described in more detail and standardized.
  • In the Materials and Methods section the authors mention 66 CBCT and 184 MDCT images
  • As above
  • In the Results section, the authors state a number of 499 models of the condylar process that were obtained, of which 367 models from fan beam tomography (FBCT) and 132 models from cone beam tomography (CBCT)

*Please consider a more organized view on the number and types of images processed

As above,  it has been described in more detail and standardized.

  1. A table should be considered for the legend of Figure 1 explained on rows 80-98.

Measurement descriptions are included in the table.

  1. Numbers 10, 11 represented on the figure don’t have a description

It has been corrected.

  1. Reference 38 noted on row 195 does not appear in the references section.

It has been corrected.

  1. The conclusion section is formulated in a vague manner; conclusions should reinforce the central idea of the paper.

The conclusion is framed in terms of points more consistent with the central idea of the paper.

Reviewer 2 Report

The study seems to be well executed and correctly planned. the authors used a correct scientific methodology. the introduction can be improved. I suggest increasing the references. I recommend to insert in the introduction: 

Manuelli, M. A peaceful man. Prog. Orthod.2012. 13(1). 1.

Rodriguez y Baena, R.; Pastorino, R.; Gherlone, E.F.; Perillo, L.; et Al. Histomorphometric Evaluation of Two Different Bone Substitutes in Sinus Augmentation Procedures: A Randomized Controlled Trial in Humans. Int. J. Oral. Maxillofac. Implants. 2017. 32(1). 188-194.

Author Response

The study seems to be well executed and correctly planned. the authors used a correct scientific methodology. the introduction can be improved. I suggest increasing the references. I recommend to insert in the introduction: 

Manuelli, M. A peaceful man. Prog. Orthod.2012. 13(1). 1.

It has been cited.

Rodriguez y Baena, R.; Pastorino, R.; Gherlone, E.F.; Perillo, L.; et Al. Histomorphometric Evaluation of Two Different Bone Substitutes in Sinus Augmentation Procedures: A Randomized Controlled Trial in Humans. Int. J. Oral. Maxillofac. Implants. 2017. 32(1). 188-194.

It has been cited.

Reviewer 3 Report

The aim set in the introduction is "Evaluation of differences in the dimensions of mandible condyle models in fan- versus cone-beam computer tomography for surgical treatment purpose". However matherials and methods fail in explaining how the research design is supposed to prove any difference between the two methods. FBCT and CBCT were taken from different subjects, measurements were arbitrarily chosen. and no reliability calculation was performed (ICC). Discussion and conclusion appear disjoined from the results and are quite vague. Why Kruskal-Wallis test was applied in a between two groups comparison is incorrect. I would rather prefer the actual value in table 1 pvalue column

Author Response

However matherials and methods fail in explaining how the research design is supposed to prove any difference between the two methods. FBCT and CBCT were taken from different subjects,

Of course the reviewer is right. For the purity of scientific assessment, this would be best. Unfortunately, the institutional bioethics committee has not approved multiple patient studies in ionizing radiation. From this comes the construction of study groups.

 measurements were arbitrarily chosen.

Measurements were chosen because of the clinical considerations of osteosynthesis in which screw lengths must be taken into account. Therefore, it is necessary to know the bone thickness at the site of planned screw insertion [thickness sigmoid nothc, Width_neck_basal, Width_head]. The second principium is the restoration of the correct height of the condyle and ramus of the mandible. Hence, measurements [ramus heigth, neck height]. Measurements Length_neck_basal, Length_neck_top, Length_neck_middle were chosen to assist the clinician in selecting the correct plate width as the fixation material.  The last important issue for treatment is correct angulation of the condyle [ High_neck_new_classification, Angle_posteriorline-notchpoint, Distance_sigmoidnotch-neck top]. The authors of this paper use the above-mentioned measurements in their daily clinical work, and it seems to them that this clinically useful selection may also be useful to other readers.

and no reliability calculation was performed (ICC).

Interclass correlation coefficient was not calculated. It is usually used to compare the strength of researchers' reliability as 2 or more observers results are considered. However, a similar statistical technique was used, based on an assessment of the ratio of between-class variance to within-class variance.

{\displaystyle F={\frac {\text{variance between treatments}}{\text{variance within treatments}}}}

Discussion and conclusion appear disjoined from the results and are quite vague.

It has been corrected

Why Kruskal-Wallis test was applied in a between two groups comparison is incorrect.

This test was used in cases comparing classes where no normal distribution was found or the variance between classes was significantly different [p<0.05].

I would rather prefer the actual value in table 1 p value column

It has been corrected

Round 2

Reviewer 1 Report

The necessary modifications have been made.

I agree with the final form of the manuscript.

Author Response

Thank you very much for your positive review.

Reviewer 3 Report

I am satisfied with the modifications performed by the reviewers in accordance to my suggestions. I furthermore suggest to add the following reference into the discussion section to broaden them DOI: 10.1038/s41598-020-68562-6;  10.3390/jcm9041159; 10.1007/s00784-019-03122-5

Author Response

The articles have been cited. Kindly thank you for your positive review.